# Therapeutic Vaccines against Hepatocellular Carcinoma in the Immune Checkpoint Inhibitor Era: Time for Neoantigens?

**DOI:** 10.3390/ijms23042022

**Published:** 2022-02-11

**Authors:** David Repáraz, Belén Aparicio, Diana Llopiz, Sandra Hervás-Stubbs, Pablo Sarobe

**Affiliations:** 1Centro de Investigación Médica Aplicada (CIMA), Universidad de Navarra, 31008 Pamplona, Spain; dreparaz@alumni.unav.es (D.R.); baparicio.1@alumni.unav.es (B.A.); diallo@unav.es (D.L.); mshervas@unav.es (S.H.-S.); 2IdiSNA, Instituto de Investigación Sanitaria de Navarra, 31008 Pamplona, Spain; 3Centro de Investigación Biomédica en Red Enfermedades Hepáticas y Digestivas CIBEREHD, 31008 Pamplona, Spain

**Keywords:** hepatocellular carcinoma, immunotherapy, vaccines, neoantigens, immune checkpoint inhibitors

## Abstract

Immune checkpoint inhibitors (ICI) have been used as immunotherapy for hepatocellular carcinoma (HCC) with promising but still limited results. Identification of immune elements in the tumor microenvironment of individual HCC patients may help to understand the correlations of responses, as well as to design personalized therapies for non-responder patients. Immune-enhancing strategies, such as vaccination, would complement ICI in those individuals with poorly infiltrated tumors. The prominent role of responses against mutated tumor antigens (neoAgs) in ICI-based therapies suggests that boosting responses against these epitopes may specifically target tumor cells. In this review we summarize clinical vaccination trials carried out in HCC, the available information on potentially immunogenic neoAgs in HCC patients, and the most recent results of neoAg-based vaccines in other tumors. Despite the low/intermediate mutational burden observed in HCC, data obtained from neoAg-based vaccines in other tumors indicate that vaccines directed against these tumor-specific antigens would complement ICI in a subset of HCC patients.

## 1. Introduction

With an incidence of 906,000 new cases and a mortality of 830,000 deaths worldwide, liver cancer represents an important medical challenge. Among the different liver tumors, hepatocellular carcinoma (HCC) emerges as the dominant form of primary liver cancer, comprising about 80% of these tumors, followed by intrahepatic cholangiocarcinoma, with 10–15% of cases [1]. Incidence and mortality rates of HCC are higher in men than in women, becoming the second tumor in terms of mortality in men. Multiple factors are implicated in the etiology of HCC, with chronic liver disease being a common factor in almost all cases. The most important include chronic viral infections caused by the hepatitis B virus (HBV) and the hepatitis C virus (HCV), alcohol intake, obesity, and diabetes, with additional factors like tobacco smoking, aflatoxins, and familial or genetic factors [2]. These risk factors vary from region to region, with HBV infection predominating in Asia, and HCV in Egypt, Japan, Western Europe, and North America, whereas obesity is becoming a relevant factor in Western societies [3]. Despite implementation of vaccination programs against HBV [4] and the use of antiviral drugs in HCV infection [5] to curb virus-induced HCC, this tumor continues to rise, mainly because of the increasing incidence of obesity, which leads to non-alcoholic fatty liver disease [6].

HCC is a complex and heterogeneous disease, resulting from the accumulation of different mutations. During the last years, several studies have addressed the molecular subtyping of HCC based on the genomic and epigenomic landscape, in association with etiological, clinical, and histological features [7,8,9]. In general, most studies classify HCC into two main types [10], one being the “proliferation class”, poorly differentiated aggressive tumors associated with HBV infection, enriched in *TP53* inactivating mutations and with activated signaling pathways like mTOR, RAS-MAP, and MET. This tumor class is subdivided into the “Wnt-TGF-β subclass”, characterized by activation of Wnt and TGF-β pathways, usually associated with an exhausted immune response [11], and the “progenitor subclass”, with upregulated expression of hepatic progenitor markers and IGF1R and AKT pathways [9]. The second, the “non-proliferation class”, contains less aggressive, more differentiated tumors associated with HCV infection and alcohol consumption. A first subclass within this group is characterized by an enrichment in mutations in *CTNNB1* and *TERT* promoter, and with an immunologically “cold” landscape, whereas a second subclass, denominated “G4”, contains tumors with upregulation of the IL-6/JAK-STAT pathway, which in some cases display an interferon-stimulated gene signature and an active immune response [11].

Although most HCC cases take place in an identifiable population, predominantly in individuals with ongoing liver disease, HCC is diagnosed in an important proportion of patients at a symptomatically advanced stage, with fewer cases detected at earlier stages. This advanced-stage diagnosis results in a different set of therapeutic options. Indeed, HCC management was improved during the last decade by adopting therapeutic strategies according to the staging system, principally the Barcelona Clinic Liver Cancer (BCLC) [12]. In this way, patients with early stage HCC, and those with small single or multinodular tumors and well-preserved liver function, are submitted to local curative therapies like tumor resection, transplantation, or ablation. In these circumstances, these therapies achieve a 5-year overall survival rate of 50–70% for resection, 70–80% for transplanted patients, and 40–70% for different ablation procedures [3]. Unfortunately, patients at intermediate stages, treated by transarterial therapies, mainly chemoembolization (TACE), have an overall survival of 20–35 months, depending on studies using TACE or combinations [13]. Finally, until a few years ago, patients with advanced HCC received targeted therapies administered systemically. Different mutations and signaling pathways have been identified in HCC patients, but in terms of targetable molecules, only 20–25% of them have known actionable mutations. Therefore, during the last decade, treatment of advanced HCC was based on the use of tyrosine kinase inhibitors (TKI). Sorafenib was the first drug approved as a first-line therapy, according to studies reporting an overall survival of 10.7 months [14], followed by lenvatinib, approved 10 years later with an overall 13.6-month survival [15]. Other agents are used as second-line therapies [16], including regorafenib and cabozantinib, with an overall survival of 10 months, and ramucirumab, which instead of blocking kinases, including those associated to VEGF signaling, is an antibody that inhibits VEGFR2, with an improvement of overall survival of 8 months.

These results suggest that HCC therapy is still far from being optimal, mainly for patients at advanced stages, who mainly have a dismal prognosis, making necessary the development of new treatments with higher response rates and prolonged overall survival. Indeed, despite the greater understanding of molecular mechanisms of HCC pathogenesis attained during the last years, staging and treatment systems such as BCLC are still based on morphological criteria to subdivide patients and direct management strategies. Therefore, there is an ongoing need to refine treatment algorithms by including molecular markers indicative of high-, intermediate-, and low-risk tumor biology.

## 2. Immunotherapy of HCC

As previously described, patients with advanced HCC have very few therapeutic options, mainly those based on the systemic administration of TKIs. Reported cases of spontaneous remissions of HCC after withdrawal of immunosuppressive agents support the idea that immunotherapy could be suitable for advanced HCC treatment [17,18]. For this reason, immunotherapeutic strategies have been tested in HCC. However, most successful results have been obtained with immune checkpoint inhibitors (ICI) previously adopted for other tumors [19,20].

T cells express co-inhibitory receptors that control the magnitude of the immune response to avoid over-activation of T cells. These molecules, also known as immune checkpoints, include CTLA-4, PD-1, TIM-3, LAG-3, and BTLA, among others. Since the discovery of these immune checkpoints and the production of monoclonal antibodies against them, the landscape of cancer therapy has changed deeply in favor of immunotherapy. Two main immune checkpoints have been considered in HCC therapy. The first is CTLA-4, targeted by antibodies tremelimumab and ipilimumab. CTLA-4 competes with CD28, expressed on the surface of lymphocytes, for binding to CD80 and CD86 in antigen presenting cells (APC), thus inducing and inhibitory signal for T cell activation. The second pathway is the one corresponding to the PD-1/PD-L1 axis, targeted by antibodies nivolumab, pembrolizumab, atezolizumab, or durvalumab, among others. PD-1, expressed by T cells, interacts with its ligand PD-L1, expressed by tumor cells and immune infiltrating cells, to suppress T cell activation through co-inhibitory signals. Different studies have been carried out with HCC patients using these antibodies as monotherapies or in combination. Moreover, they have been combined with other drugs approved for HCC, such as TKI, and, more recently, the combination of anti-PD-L1 antibody atezolizumab and the anti-VEGF bevacizumab has been approved for first-line therapy. In general, overall response rates (ORR) are in the range of 15–30%, from monotherapies to the most efficient combination therapies (Table 1).

However, from these results, as occurs in other tumors, it is clear there is still a proportion of non-responder patients who should be treated with new protocols, either reinforcing these therapies by combined regimens, or by designing alternative therapies targeting other immune-related molecules. In this respect, evaluation of the immune landscape observed in each patient may aid to rationally design the most appropriate therapies.

## 3. Tumor Microenvironment and Immune Response in Liver Cancer

The tumor-immune microenvironment may define the fate of immune-based therapies. In this regard, both the healthy and cancerous liver have special features. The predetermined immune state of the liver is anti-inflammatory and tolerogenic, which is crucial to establish immune tolerance against innocuous molecules such as food antigens (Figure 1A) [30]. However, the liver also plays an important role in the defense against pathogens and under appropriate conditions can induce a robust immune response. These opposing features determine the tumor microenvironment in liver cancer.

### 3.1. Antigens Associated with Liver Cancer

As mentioned, HCC arises on the background of a chronically inflamed liver. This chronic inflammation has important implications: on the one hand, it induces the accumulation of reactive oxygen species, which in turn generate epigenetic changes and chromosomal instability promoting tumor initiation. Moreover, it can also promote the appearance of tumor-associated antigens (TAA), either by deregulating the expression of oncofetal or testicular cancer antigens [31]. Indeed, cellular response has been detected against alpha-fetoprotein (AFP), glypican-3 (GPC-3), melanoma-associated genes (MAGE)-1, 3, and 10, synovial sarcoma X (SSX)-2, and New York-esophageal squamous cell carcinoma-1 (NY-ESO-1) in blood, as well as in the tumors of HCC patients [31,32,33,34]. In addition, humoral response against the NY-ESO-1 has also been detected in patients with HCC tumors that express this antigen [35]. The presence of this tumor-specific T cell response correlates with patient survival [31].

Along with shared antigens, proteins resulting from somatic mutations can generate tumor-specific neoantigens (neoAgs) that markedly increase tumor immunogenicity [36]. Some of these neoAgs are the products of driver mutations and are shared by several types of tumors and patients (e.g., *TP53*), while the majority are private neoepitopes resulting from passenger somatic mutations. CD4 T cells specific for a non-synonymous mutation have been identified in a metastatic cholangiocarcinoma patient presenting 26 predicted neoAgs [37]. This finding suggests that somatic mutations may also be responsible for the spontaneous immune response detected in liver cancer and that they could be appropriate target antigens for immunotherapy.

### 3.2. Effector Immune Cells in the TME of Liver Cancer

The decreased number and impaired effector functions of tumor-specific T cells are related to tumor progression. In the case of HCC, where circulating and tumor-infiltrating CD8 and CD4 T cells are significantly increased in the early stage of the disease, their numbers decrease in later stages [31,38]. On the other hand, tumor-infiltrating CD8 T cells are not capable of responding to tumor-antigen stimuli, while their counterparts in peripheral blood do, indicating that tumor-infiltrating T lymphocytes (TILs) are exhausted in HCC patients [31].

NK cells play a central role in the immune control of HCC [39]. Several mechanisms seem to be involved in the lack of antitumor control by NK cells: NK cell receptor (KIR) polymorphism [40], expression of inhibitory receptors (such as NKG2A) [41], suppression of NK by myeloid-derived suppressor cells (MDSC) [42], and the appearance of dysfunctional populations of CD11b-CD27-NK cells [43].

### 3.3. Suppressor Cells in the TME of Liver Cancer

The tolerogenic nature of the liver depends on different subsets of hepatic nonparenchymal cells, with antigen presenting functions, such as Kupffer cells (KCs), hepatic stellate cells (HSCs), liver sinusoidal endothelial cells (LSECs), dendritic cells (DC), and myeloid derived suppressor cells (MDSC). KCs, liver-resident macrophages, express inhibitory molecules, such as IL-10, prostaglandins, IDO, and PD-L1, and exhibit a low expression of costimulatory molecules [44]. In HCC, KC in the peritumoral margin express higher levels of PD-L1 compared to non-tumorous liver, thus inhibiting CD8+ T cell effector functions [45]. They also promote the activation of Tregs [46]. LSECs regulate the effector immune response in the liver [47] through expression of high levels of PD-L1 and low levels of costimulatory molecules CD80 and CD86, as well as the induction of Tregs in a TGF-β-dependent manner. LSEC also reduce the ability of dendritic cells (DC) to activate T cells [48]. In fact, hepatic DCs contribute to the tolerogenic microenvironment of the liver by expressing low MHC II and co-stimulatory molecule levels and producing anti-inflammatory molecules like prostaglandin E2 (PGE2), which in turn increase the secretion of IL-10 and induce Tregs cells [49]. HSCs release the hepatocyte growth factor, playing a role in HCC progression and promoting MDSC [50] and Treg accumulation [51]. Moreover, HSCs also induce T cell apoptosis through PD-L1 expression [52].

MDSCs are a heterogeneous cell population of immature myeloid cells that exert pro-tumor functions through different mechanisms, such as the production of cytokines and other molecules that favor the survival and propagation of tumor cells, the formation of new blood vessels, and the inhibition of T [53] and NK cells [42]. Immature MDSCs are recruited by cytokines and chemokines secreted by tumor cells. Tumor cells prevent the differentiation of these cells to macrophages, remaining in an immature state that contributes to creating an immunotolerant environment [54]. A specific MDSC subset (CD14pos HLA-DRneg/low) found in the tumor tissue and peripheral blood of patients with HCC is characterized by the production of IL-10 and TGF-β, which induce Tregs [55] and are associated with tumor progression [56].

Tregs are CD4 T lymphocytes that express CD25 and FoxP3, as well as high levels of CTLA-4. The mechanisms used by Tregs to exert their inhibitory functions are very diverse and include the production of inhibitory cytokines, such as TGF-β and IL-10, the depletion of IL-2 by the IL-2 receptor (CD25), and the sequestration of CD80 and CD86 on APC by CTLA-4 [57]. Chemokines such as CCL20 [58] and CCL22 [59] mediate the recruitment of Tregs in HCC. In HCC patients, FoxP3^+^ Tregs are increased both in the tumor [60] and in the periphery [46], and their presence in the tumor correlates with the presence of tumor macrophages [61].

Several other immune or stromal cell types cooperate for the generation of an immunosuppressive tumor microenvironment: Th2-secreting invariant natural killer T (iNKT), enriched in intrahepatic malignant tumors [62] and predictive of shorter time to recurrence [63]; regulatory B cells, expressing high levels of PD-1 and with the capacity to suppress anti-tumor T cell response and promote disease progression [64]; Th17 CD4 T cells, present at a high frequency in peripheral blood from patients with HCC and responsible for impairing CD8+ T cell effector functions [65]; TIE2+ monocytes, related to angiogenesis and poor prognosis [66]; a population of CD14+ DCs expressing high levels of CTLA-4 and PD-1 and producing IL-10 and indoleamine-2,3-dioxygenase (IDO) [67]; neutrophils responsible for macrophage and Tregs recruitment, which foster tumor progression and resistance to sorafenib [68]; and tumor-associated fibroblasts (TAFs), originated either from portal fibroblasts or from HSCs, which support tumor progression, inhibit NK-cell function, and induce MDSC differentiation, thus impairing anti-tumor immunity [69].

In summary, the liver has a plethora of cell subsets with tolerogenic functions over-represented in HCC patients, contributing to impair the antitumor response (Figure 1B).

### 3.4. Heterogeneity of TME in HCC

The cellular composition of TME varies among HCC patients. Considering the level of lymphocyte infiltration, human tumors have been categorized as inflamed, immune desert, or immune-excluded phenotypes [70]. Within the HCC, a series of immune subclasses has also been defined. Llovet et al. separately analyzed gene expression profiles from tumor, stromal, and immune cells from 956 HCC using a non-negative matrix factorization algorithm [11]. They found that approximately 25% of HCC express PD-1 and PD-L1 and markers of cytolytic activity and tertiary lymphoid structures. This group, referred as the “immune class”, associates with a better median overall survival. Further stratification identified two subtypes within the immune class, characterized by markers of an adaptive T cell response or exhausted immune response. The “active immune” sub-class displays signatures related to effector T cells, whereas the “exhausted immune” sub-class exhibits enrichment in genes regulated by TGF-β1 and in those characteristic of immunosuppressive macrophages. A third immunological class has been described, which is characterized by presenting an immunosuppressive signature in the tissues surrounding the tumors, but little immune gene expression in the tumor core. This class has been called “immune excluded”, appears in ~25% of patients with HCC, and is associated with a poor prognosis. Interestingly, this class overlaps with a subset of tumors with an activated WNT-β-catenin pathway [71,72]. Using multiplex immunohistochemistry, Kurebayashi et al. classified HCC into three immune subtypes: “immune-high”, “immune-mid”, and “immune-low” [73]. Consistent with the “immune class” of Llovet et al., the “immune-high” subtype is enriched in T cells and B/plasma cells and associates with a good prognosis. By integrating multiomic analysis, Zang et al. expanded these observations and identified three distinctive HCC subtypes with immunocompetent, immunodeficient, and immunosuppressive features [74]. The “immunocompetent subtype”, characterized as CD45high FOXP3low by immunohistochemistry, has high infiltration of γδ T cells. In addition, the immunosuppressive subtype, characterized by high FOXP3 and CD45 staining, has a high frequency of Tregs, B lymphocytes, and macrophages, as well as expression of immunosuppressive molecules, such as PD-1, PD-L1, CTLA-4, VEGF, and TGF-β. Finally, as expected, the CD45low subtype exhibits scant lymphocytic infiltration. In summary, these studies highlight the marked TME heterogeneity in HCC. This heterogeneity may reflect the different mechanisms of immune response and escape that the tumor has experienced during its evolution. Stratification of patients on the basis of their TME class would help to identify potential immunotherapeutic targets.

## 4. Vaccines against HCC

Immune enhancing strategies would be of help to those patients with tumors lacking a lymphocytic infiltrate amenable to treatment with ICI. Among these strategies, vaccination is one of the first immunotherapeutic approaches used in HCC. Current treatments for advanced stage HCC still have limited efficacy and cannot prevent the high recurrence rate. Indeed, in the past, vaccines emerged like possible tools to tackle this issue, willing to improve clinical outcomes when used in combination with already approved systemic treatments. Nevertheless, few trials have been conducted to date, all are phase I or II trials, most of them are quite old, and although they were proved to be safe and have immunologic effects, they have only provided underwhelming/poor clinical results/efficacy [75,76] (Table 1).

HCC vaccination strategies performed hitherto can be classified as peptide-based or DC-based vaccines (Table 2). The latter can also be subclassified into peptide-loaded DCs and tumor lysate-pulsed DCs. The main antigens used for peptide-based vaccines in HCC include epitopes from oncofetal antigen alphafetoprotein (AFP) [77,78] and glypican 3 (GPC-3) [79], and the human telomerase reverse transcriptase (hTERT) peptide GV1001 [80]. DCs can also be loaded with peptides and clinical trials have been done using peptides from AFP [78] and AFP combined with MAGE-1 and GPC3 [81]. Clinical trials using both autologous tumor lysates [82] and HepG2 (hepatoma cell line) lysates [83] have been used too.

All these vaccination strategies have been demonstrated to be safe, and most of them induced antigen-specific responses without toxic or autoimmune reactions, although clinical responses were poor. This limited efficacy could be attributed to the diverse features of HCC tumors and vaccine design, or even the combination of both. TAA-based vaccines are not completely tumor specific; therefore, they are subjected to tolerance mechanisms, reflected in a scarcity of highly reactive clones against them [87], and resulting thus in most cases in responses without sufficient potency to overcome tumor progression. Moreover, as previously described, the immunosuppressive HCC environment, a clear pronounced reflection of the intrinsic liver environment, is not propitious for immune responses [20]. Moreover, TAAs are not ubiquitously expressed in HCC tumors and their number is limited, which can lead to immune escape by Ag loss. The different vaccine modalities used in HCC patients have their own advantages and disadvantages (summarized in Table 3) with regard to their production, the antigenic repertoire, and the range of patients to be treated.

Not all vaccination approaches have yet been exploited in HCC clinical trials. Vaccines may be improved by modifying the vaccine platform or by including new tumor antigens. The first includes strategies such as in vivo DC-targeted vaccination [88] and tumor cell fusion [89]. Targeted vaccines are based on the linkage of the antigen to antibodies, ligands, or viruses, which in theory reduces potential adverse effects by preventing non-target cell Ag delivery. We recently demonstrated in a preclinical model that this strategy improves the therapeutic efficacy of ICI when added in a combined treatment [90]. With regard to a wider antigenic repertoire, other TAAs with the potential to be included in future clinical trials are NY-ESO-1 [91], WT-1, ROBO1, and FOXM1 [92]. Finally, neoAgs, because of their tumor specificity and their potential higher immunogenicity, should be considered for future HCC antitumor vaccination approaches [76,93].

## 5. Neoantigens as New Targets for Vaccination

NeoAgs are new protein sequences resulting from mutations appearing in tumor cells. The vast majority of these mutations are found in exons [94], but there are neoAgs derived from mutations in adjacent intron sequences [95]. NeoAgs derive from genetic alterations that are essentially specific for each patient (unique), and are considered as “passengers”, as they normally do not play a key role in the cellular transformation [96]. Since they are highly tumor-specific, they can be considered as tumor-specific antigens (TSA). Interestingly, they are not subjected to central tolerance mechanisms, which confers on them a high antigenicity [97,98], and makes them interesting molecules as potential response biomarkers and as vaccine-based immunotherapy targets. As a consequence, the therapeutic focus directed at these TSA is personalized [99,100,101,102].

### 5.1. Mutations Involved in neoAg Generation

The most important source of neoAgs are single nucleotide variants (SNVs), also known as non-synonymous “occasional” mutations that produce substitutions of amino acids. SNVs have been an important focus of interest since tumor mutational burden (TMB) of non-synonymous mutations was correlated with the response to checkpoint inhibitors [103,104,105]. The enhanced immunogenicity of the new epitopes deriving from SNVs is due to the new amino acid, yielding an improved contact with the T cell receptor (TCR) or a new epitope with enhanced anchoring and presentation capacities by MHC molecules [106].

In addition to SNVs, nucleotide insertions and deletions (“indels”) in the coding regions originate changes in the reading frame, which represent another source of neoAgs. Tumors with better responses to checkpoint inhibitors have a higher prevalence of mutations caused by “indels” [107]. Indeed, it has been reported that 10% of the MHC-I-presented ligands were peptides caused by “indels” [108].

Chromosomic translocations can also lead to the creation of neoepitopes that bear a mutation in the breaking point, and for that they represent another source of potential neoAgs, as demonstrated in different tumors [109]. However, the lack of natural processing/presentation of the mutated ligand could be the cause of the low response rate to vaccines based on these neoAgs [110].

Altered processes in tumor cells, such as post-translational modifications (phosphorylation and deamination) or alternative splicing, may also originate new tumor-specific epitopes [111,112]. It has been recently observed that new epitopes originated by alternative splicing substantially contribute to the immunopeptidome [113]. Although neoAgs resulting from these processes are an interesting target for immunotherapy, there are no reliable prediction algorithms for the identification of these neoAgs.

### 5.2. Factors Determining neoAg Immunogenicity

Although mutations can originate new sequences, they do not always result in immunogenic neoAgs. In the case of T cell responses, the mutated sequences must be expressed and processed by tumor cells and processed by APC. The lack of appropriate cleavage sites generating the correct neoepitope peptide would prevent neoAg presentation by MHC molecules and TCR recognition. For this to occur, these peptides also need sufficient affinity for MHC binding. Once presented, dissimilarity between the wild type and the mutated sequences would facilitate recognition by the available TCR repertoire resulting after thymic-negative selection. A recent study indicated that approximately 0.5% of mutated peptides expressed by the tumor are recognized by TILs [96].

Clonality is another relevant feature for neoAg properties. There are clonal neoAgs, arising soon and developing earlier during the transformation process, and subclonal neoAgs, those that appear later and only in a subset of cells as the tumor evolves. A recent study reported that patients with a higher rate of clonal neoAgs survived longer and had a higher response rate to ICI in comparison with patients with a higher subclonal repertoire [114]. In summary, not only is the amount important, but so is the presence of neoAgs in the higher number of tumor cells.

Finally, the neoAg repertoire expressed by the tumor is also affected by its interaction with the immune system. This phenomenon, known as “cancer immunoediting”, is the result of the stochastic nature of the tumor-specific mutations and the selection processes exerted by immune recognition of these antigens. T cells play a key role in modulating tumor antigenicity by the immune-selection process, destroying tumor cells that express highly antigenic TSA and sparing cells with weaker (less immunogenic) neoAgs. These mechanisms are present during immunotherapy treatments; thus, immunotherapy can re-edit the tumors [115].

### 5.3. neoAg-Based Vaccines

In addition to its association with treatment response, neoAg identification is helping to develop personalized neoAg-based therapies. This includes (i) neoAg-based vaccines and (ii) adoptive T-cell therapy. Regarding vaccines, this strategy was first developed in preclinical models, using whole-exome sequencing and transcriptome sequencing (RNAseq) technologies to identify SNVs that were expressed in murine tumor cell lines. Mutated peptide sequences were filtered with MHC-binding prediction algorithms [116] or by mass spectrometry [117]. Selected peptides were synthetized and tested in vivo in immunization assays, demonstrating efficacy in the induction of neoAg-specific T cells and in the delay of tumor growth after vaccination [116,117,118].

After proof-of-concept preclinical experiments, this approach is being developed with cancer patients. In fact, different vaccination strategies, such as peptide-based, RNA multiepitope, and DC vaccines, have been tested in stage III and IV melanoma patients [119,120,121,122].

## 6. Mutations in HCC as Elements for neoAg-Based Vaccines

Compared to the high levels of TMB found in skin and lung cancers (hypermutated cancers), or to the lowest levels characteristic of leukemias and pediatric tumors [123], HCC is considered as a low to moderate mutated tumor. Its TMB ranges from 2 to 5 somatic mutations per megabase (Mb), reflecting in approximately 60 non-synonymous mutations within the exomic regions [124,125,126]. HCC mutations are not evenly displayed all over the tumor cell genome; there are mutational hot spots such as *CTNNB1*, *TP53*, *NBPF1*, *MUC4*, *MUC16*, *ALB*, *ARID1A*, *AXIN1*, *APOB*, and *ALB* [10,72,127]. Nevertheless, the SNVs present in these genes are rather unique among patients [127], even if it seems that there is a predominance of (C > T), (C > A), (T > C), and (T > G) substitutions [125,128]. Depending on the etiology of the HCC, along with other patient-intrinsic factors, the set of mutated genes can change too (mutational signature), so this is an important feature to consider for improving the identification pipeline and the selection of potential neoAgs for vaccination [125,129].

By analyzing the presence of mutations, recent studies have predicted an average of 9–15 neoAgs in HCC patients [130,131]. However, these initial studies did not confirm immunogenicity, and most of these predicted neoAgs are restricted just to in silico analyses. Proteomic studies using mass spectrometry to detect HLA-bound neoAg peptides have failed to confirm the presence of these epitopes in HCC samples, but the complexity of these techniques should be considered as a limitation, inter alia, because of the outnumbering amount of self-antigens in tumor cells [127,132]. Most confirmation assays are restricted to recognition by T cells. Indeed, in addition to our recent study showing that TILs are able to recognize predicted neoAgs of autologous tumors [133], neoAg-reactive T cells have been successfully isolated from tumors and peripheral blood in HCC patients, suggesting the ability of current neoAg identification pipelines to identify these epitopes [134].

While it is true that, generally, TMB correlates with prognosis, survival, or even response rate to ICI blockade [135], this does not seem to be true for HCC, and it may not depend just on neoantigen quantity, but also on their quality [136]. In this respect, it has been reported that neoAgs derived from *TP53* mutations are associated with better prognosis, lymphocyte infiltration, and even cytolytic activity [137]. Moreover, a recently published study reports that overall survival may correlate better with the amount of high affinity neoAgs than with total TMB [134]. In this last case, it was also found that the presence of increased numbers of high-affinity neoAgs was associated with a better prognosis after anti-PD-1 therapy, suggesting that it could be also useful to predict response to ICI.

As previously described, monotherapies based on vaccines have not been successful in HCC, presumably because of the immunosuppressive microenvironment observed in HCC. Despite the high immunogenicity and specificity of neoAgs, vaccines based on these antigens may behave similarly. Therefore, future strategies should consider the combination of vaccines with immune-stimulating agents and blockade of immune checkpoints to reverse the immunosuppressive environment [129]. A recent vaccination study with personalized neoAgs in 10 HCC patients has demonstrated vaccine safety and immunogenicity, but the low patient number does not allow us to reach solid conclusions on clinical activity [126]. In the same line, and with the aim of improving efficacy, a recent phase I/II clinical trial has assessed the safety of a DNA plasmid-based vaccine encoding patient-specific neoAgs (GNOS-PV02) in combination with IL-12-producing plasmid (INO-9012) and pembrolizumab (PD-1), yielding promising results [138].

## 7. neoAg-Based Vaccines as Combinatorial Partners with Immune Checkpoint Inhibitors

Since the identification of cancer antigens, the development of vaccines to prevent or treat different types of tumors has been a continuous challenge for immunologists. However, most clinical vaccination trials have not yielded the expected therapeutic results, and therefore only a few vaccines are approved in the context of cancer. In addition to prophylactic vaccines against HBV and human papilloma virus (HPV), which prevent the development of liver cancer and of several HPV-associated tumors, the only therapeutic vaccine available is Provenge for prostate cancer [139]. However, with the identification of immune checkpoints, and the characterization of the immunosuppressive environment observed in the tumor, it has become evident that, even in those vaccines with strong immunogenic properties, there are elements beyond vaccine immunogenicity that prevent tumor rejection. Therefore, new strategies should combine priming strategies with inhibition of immunoregulatory elements.

In this regard, most studies have combined vaccination based on the administration of TAA and ICI. Vaccines based on cancer-related viral antigens have been evaluated in combination with anti-PD-1 in recurrent HPV-driven cancer. Thus, the combined therapy of the vaccine ISA 101, a synthetic long-peptide from HPV-16 and nivolumab, rendered an overall response rate of 33% and a median overall survival of 17.5 months [140], well above the results obtained with checkpoint blockade approved in the second line, achieved an overall response rate of 14.3%. Similarly, the administration of GX-188E DNA vaccine (encoding HPV-16 and HPV-18 E6 and E79), plus pembrolizumab, to patients with recurrent or advanced cervical cancer resulted in an overall response rate of 42% [141].

There are also ongoing clinical trials, the results of which have not yet been published, combining vaccines with ICI. The combination of a pTVG-HP DNA vaccine-encoding prostatic acid phosphatase (PAP) with pembrolizumab is being tested in patients with metastatic prostate cancer (NCT02499835). Also, in glioblastoma patients, there are combinations that include the ATL-DC vaccine (autologous dendritic cells pulsed with tumor lysate) and pembrolizumab (NCT04201873) or the IMA950 peptide vaccine (composed of peptides eluted from the surface of glioblastoma samples) in combination with poly-ICLC and pembrolizumab (NCT03665545). Moreover, another study in patients with selected advanced cancers (metastatic ovarian cancer, acute myelogenous leukemia, colorectal cancer, triple-negative breast cancer, and small-cell lung cancer) is testing the efficacy of the Galinpepimut-S vaccine (which contains peptides of the WT1 protein) combined with pembrolizumab (NCT03761914).

Regarding other tumor antigens, it is known that the therapeutic effect of ICIs is partly mediated by responses against neoAgs, and at the same time response rates to these therapies is associated with TMB (a putative correlate of the number of neoAgs and tumor immunogenicity). Therefore, it seems evident that to increase the response rate to ICI, in addition to vaccines promoting responses against TAA, activation of neoAg-specific immunity would result in more inflamed tumors, potentially amenable to treatment with these antibodies. As described above, several neoAgs-based vaccines have been tested in clinical trials. Several clinical trials have demonstrated the safety and efficacy of personalized neoAg vaccines [120,121,122,142,143]. In some of these studies, selected patients were also treated with CTLA-4 [122] or PD-1-blocking antibodies [120,121]. However, despite some complete responses observed in some of them, the low number of patients receiving the combined therapy makes it difficult to draw solid conclusions about the potency of this strategy. More recently, a clinical trial was designed to test the combined effect of neoAg vaccine NEO PV 01 and anti-PD-1 antibody nivolumab in patients with melanoma, bladder cancer, and NSCLC. Immune analyses revealed that there were few preexisting neoAg responses, but after vaccination, neoAg-specific functional T cells were induced. In addition, epitope spreading to neoAgs not included in the vaccine was detected post-vaccination. An ORR of 59% was observed in melanoma, 39% in NSCLC lung cancer, and 27% in bladder cancer [144]. These results encourage the designing of combinatorial treatment with personalized antitumor vaccines and ICI in other tumors, including HCC.

Moreover, other clinical trials are testing neoAg vaccines in combination with other immunotherapies, mainly with checkpoint inhibitors, in different solid tumors. In this context, ICI used are antibodies against molecules of the PD-1/PDL-1 axis (NCT02287428, NCT03359239 and NCT04397003), or against PD-1/PD-L1 and CTLA-4 (NCT04117087, NCT03606967).

## 8. Concluding Remarks and Perspectives

Cancer vaccines aim to educate the immune system to recognize and kill tumor cells. Vaccines have been shown to be effective in preventing diseases caused by viruses and bacteria, but their success in treating HCC and other tumors has been very limited, if not null. Traditionally, HCC vaccines have been directed at TAA. One of the reasons for the low efficacy of cancer vaccines may be an immunological tolerance to self-antigens (like TAA), which prevents the induction of a powerful antitumor immune response. Unlike TAA, neoAgs are absent from healthy cells and are distinguished from germ lines, making them an ideal target for antitumor vaccine treatments. These types of vaccines represent, in theory, several advantages over other forms of immunotherapy, including fewer side effects caused by the tumor specificity of neoAgs, and the possibility of better long-term tumor control based on the induction of memory T cells. However, each individual’s tumor is unique and has its own distinctive mutations, making neoAg vaccines personalized treatments. In addition, neoAg-based vaccines require next-generation whole-exome sequencing and the use of bioinformatics and artificial intelligence algorithms for the identification and prediction of neoepitopes with high immunogenic potential. All this increases the cost of these vaccines. On the other hand, it is still necessary to identify which are the relevant neoAgs for cancer vaccines. Certain works point to clonal mutations as the most determining. In theoretical terms, the ideal neoAg should arise from mutations in driver genes, which would reduce the risk of immune escape. However, the majority of identified neoAgs are transient mutations in irrelevant genes. The administration route of a therapeutic tumor vaccine is also a critical factor in inducing antitumor activity. Intramuscular injection is the most commonly used administration route, along with the subcutaneous route, because of their easy access and safety. However, the homing behavior of T cells depends on the immunization route, with effector cells elicited by a particular immunization route preferentially homing in on tumors present at proximal sites in the body. Intratumoral vaccination has emerged as an administration route that is superior to intramuscular and subcutaneous delivery and that has the potential to reprogram the tumor microenvironment. It would be interesting to study whether intratumoral vaccination improves the efficacy of neoantigen vaccines in HCC patients.

The immunosuppressive microenvironment of liver cancer and even of a normal liver may also pose an obstacle to the success of cancer vaccines. The combination of these with other forms of immunotherapy, such as ICI and antiangiogenic agents, will be necessary to promote their action. The possibility of combining vaccines with other forms of treatment beyond immunotherapy also has an interesting potential. Thus, the combination with radioembolization and chemoembolization could improve the response of T cells through different mechanisms, such as the reshaping of the immunosuppressive tumor microenvironment, an increased T-cell trafficking to the tumor, and the release of antigens that would enhance the action of T lymphocytes induced by the vaccine and favor the antigen spreading. The possibility of using vaccines in combination with surgery, particularly in the adjuvant setting, could extend disease-free time.

Despite the recent boom in neoAg vaccines, their potential for treating HCC remains to be demonstrated. The use of neoantigen-based vaccines and their potential combination with other therapies for the treatment of HCC would require a multidisciplinary team that includes hepatologists, pathologists, biologists, and sequencing and bioinformatics services. The participation of all these experts will be necessary to make the best therapeutic decisions (patient to be treated, type, number, and length of neoantigens to be used, administration route, type of combinations) and thus have greater guarantees of success. The logistical coordination of this team is essential and the time from the collection of the tumor sample to the administration of the vaccine is critical.

## Figures and Tables

**Figure 1 ijms-23-02022-f001:**
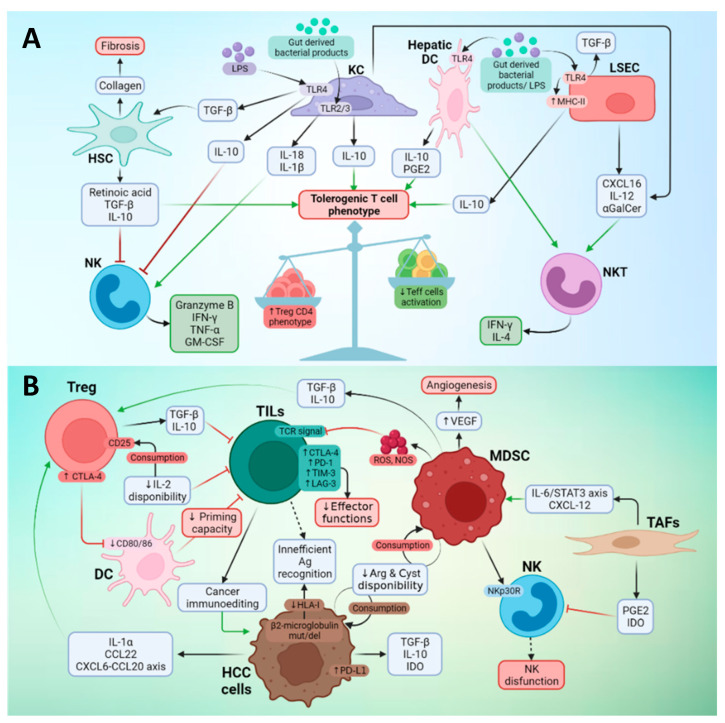
Immune landscape in a healthy liver and in HCC. (**A**) Immune status in liver homeostasis. The liver is constantly exposed to antigens coming from the digestive tract, such as bacterial-derived products and diet nutrients. In this scenario, the liver has different cell subsets that promote a tolerogenic state, formed by the KCs (liver-resident macrophages), LSEC, HSC, and hepatic DCs. These cells are exposed to gut-derived antigens and are stimulated to liberate different soluble factors that (i) favor a tolerogenic phenotype in T cells and (ii) stimulate the innate cell subset, including NKs and NKTs. (**B**) Immune microenvironment in HCC. Tumor cells modulate the immune microenvironment by releasing anti-inflammatory cytokines and altering expression of antigen-presenting molecules. Tumor-associated fibrosis favors recruitment of MDSCs, which liberate proangiogenic VEGF and immunosuppressive TGF-β and IL-10. This immunosuppressive situation (i) promotes an exhausted T cell phenotype in TILs by enhancing inhibitory checkpoint expression and consequently reducing their effector functions, (ii) favors CD4 regulatory T cell activity, and (iii) promotes NK dysfunction. This figure was created using BioRender.

**Table 1 ijms-23-02022-t001:** Immunotherapy with ICI in HCC (clinical trial with reported results).

Treatment	Patients (n)	Setting	ORR% (CRR%)	mOS (Months)
Nivolumab [21]	371	1 L	15(4)	16.4
Pembrolizumab [22]	278	2 L	18 (2)	13.9
Camrelizumab [23]	217	2 L	15 (0)	13.8
Durvalumab [24]	104	1 L/2 L	11 (0)	13.6
Tremelimumab [24]	69	1 L/2 L	7 (0)	15.1
Atezolizumab [25]	59	1 L	17 (5)	NA
Durvalumab and Tremelimumab (different doses) [24]	159	1 L/2 L	9.5–24 (1–2) NA	11.3–18.7
Nivolumab and Ipilimumab (different doses) [26]	148	2 L	31–32 (0–8)	12.5–22.8
Pembrolizumab and Levantinib [27]	100	1 L	36 (1)	22
Nivolumab and Cabozantinib [28]	36	1 L/2 L	14 (3)	21.5
Nivolumab, Ipilimumab and Cabozantinib [28]	35	1 L/2 L	31 (6)	NE
Atezolizumab and Bevacizumab [29]	336	1 L	27 (6)	NE

1 L, first-line therapy; 2 L second-line therapy; CRR, complete response rate; mOS, median overall survival; NA, not available; NE, not evaluable; ORR, overall response rate.

**Table 2 ijms-23-02022-t002:** Vaccination clinical trials in HCC with reported results.

Vaccine	Patient Inclusion Criteria	Patients (n)	Immune Response (%)	Clinical Response CR/PR/SD/PD	Observations
AFP HLA-A*02 restricted peptides+IFA	AFP+ tumors from (stage IV patients) [77]	6	66	0/0/0/6	Increased CTL response
AFP HLA-A*24:02 restricted peptides+IFA	Stage B/C tumors [84]	15	33	1/0/8/6	Increased CTL response
GPC3 HLA-A*24:02 and HLA-A*02- restricted peptides+IFA	Advanced or metastatic HCC [85]	33	91	0/1/19/13	Antitumor efficacy
GPC3 HLA-A*24:02 and HLA-A*02- restricted peptides+IFA	Patients undergone curative resection Vaccines as Adjuvant therapy [79]	41	85	Not applicable	Improved recurrence rate
Gv1001 peptide + GM-CSF + cyclophosphamide	Advanced-stage HCC with no previous antitumor treatment [80]	37	0	0/0/17/20	None clinical nor detected immunological response
DCs pulsed with AFP HLA-A*02 restricted peptides	Stage IV patients pretreated with surgery and/or chemotherapy [78]	10	60	0/1/0/9	No objective clinical responses
DCs pulsed with fused recombinant proteins (AFP, MAGE-1 and GPC-3)	After surgical resection and locoregional therapy [81]	12	92	Not applicable	Trend to improved survival
DCs pulsed with autologous tumor lysate	Advanced HCC [82]	31	0 *	0/4/17/10	Improved survival
DCs pulsed with autologous tumor lysate	Unresectable HCC [86]	8	62	0/0/4/3	Immune response generation
DCs pulsed with hepatoma cell-line (HEP-G2) lysate	No other therapeutic option [83]	35	11.4	0/1/6/18	Evidence of antitumor efficacy

Clinical trials with no published results have been excluded. CR (complete response), PR (partial response), SD (stable disease), PD (progressive disease). * immunologic response measurement was not appropriate.

**Table 3 ijms-23-02022-t003:** Advantages and disadvantages of vaccination strategies used in HCC.

Vaccine Type		Advantages	Disadvantages
Peptides		Easy preparation Known target Ag	Adjuvants required HLA restricted Limited Ag repertoire
DCs		Does not require adjuvants	Labor-intensive in CMCF Individualized manufacture
	Peptide pulsed	Known target Ag	HLA restricted Limited Ag repertoire
	Protein pulsed	Not HLA restricted Known target Ag	Protein synthesis is more challenging Limited Ag repertoire
	Tumor lysate pulsed	Not HLA restricted Full Ag repertoire available	Tumor samples not always available Predominance of self-antigens that may eclipse tumor antigens
	Cell line pulsed	Not HLA restricted Unlimited Ag source	Ag repertoire may not coincide Responses against cell line-specific Ags

CMCF: Cell Manipulation Core Facility.

## Data Availability

Not applicable.

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
