# Peer review of "Therapeutic Vaccines against Hepatocellular Carcinoma in the Immune Checkpoint Inhibitor Era: Time for Neoantigens?"

_ijms, 2022, doi:10.3390/ijms23042022_

Round 1
Reviewer 1 Report
The paper is well written and informative.
383 Although mutations can originate new sequences, they not always result in immu-nogenic neoAgs
Comments: It should read "Although mutations can originate new sequences, they do not always result in immu-nogenic neoAgs''.
I recommend the paper for publication.
Author Response
The paper is well written and informative.
383 Although mutations can originate new sequences, they not always result in immu-nogenic neoAgs
Comments: It should read "Although mutations can originate new sequences, they do not always result in immunogenic neoAgs''.
I recommend the paper for publication.
We thank Reviewer 1 for the comments and suggested changes. Accordingly, we have modified line 383 indicating "they do not always result....."
Reviewer 2 Report
Thank you for the opportunity to review this review manuscript on vaccines against HCC. While the manuscript appears rather comprehensive, it is also very dense and at times difficult to follow for an uninitiated reader. If the authors want their review to appeal to a broader readership, they need to reduce the text length and summarize more information in well-organized tables, for example. A table summarizing results of studies using ICI in HCC would be helpful, for example, as would a table describing potential advantages/disadvantages of different vaccine strategies.
In Table 1, “Patient Data” would better be described as “Patient Inclusion Criteria”. Also, please separate data listed the third column “Clinical Response” into individual columns, as the information contained herein is confusing in the manner in which it is currently presented.
In spite of greater understanding of molecular mechanisms of HCC pathogenesis, staging and treatment systems such as BCLC still based on morphological criteria to subdivide patients and direct management strategies. I would appreciate it if the authors were to provide some comment on the fact that there is ongoing need to refine treatment algorithms by including molecular markers indicative of high-, intermediate-, and low(er)-risk tumor biology, for example.
In Figure 1, the authors should differentiate “non-tumoral” liver from HCC, as the liver in the image is clearly not healthy. If they are truly aiming to differentiate healthy liver from a liver with chronic liver disease and HCC, then they should modify the image accordingly. Also, image “A” really adds nothing to the understanding of the readership (ball of cells representing a tumor, superimposed on the image of a “healthy liver”) and can be removed from the manuscript.
Author Response
Thank you for the opportunity to review this review manuscript on vaccines against HCC. While the manuscript appears rather comprehensive, it is also very dense and at times difficult to follow for an uninitiated reader. If the authors want their review to appeal to a broader readership, they need to reduce the text length and summarize more information in well-organized tables, for example. A table summarizing results of studies using ICI in HCC would be helpful, for example, as would a table describing potential advantages/disadvantages of different vaccine strategies.
We agree with the reviewer that the manuscript can be better presented by reducing the text length and by using well-organized tables. Accordingly, we have eliminated or reduced in the text some paragraphs or sections and they have been substituted by tables:
- In Section 2 “Immunotherapy of HCC” we have eliminated the paragraphs with detailed description and results obtained in seminal clinical trials. Instead, we summarized the main types of therapies (monotherapies and combination therapies of ICIs and other drugs) and we added Table 1, which summarizes clinical trials with reported data.
- The text corresponding to advantages/disadvantages of the different vaccine modalities used in HCC has been deleted and it is now presented in the new Table 3, which includes this information.
In Table 1, “Patient Data” would better be described as “Patient Inclusion Criteria”. Also, please separate data listed the third column “Clinical Response” into individual columns, as the information contained herein is confusing in the manner in which it is currently presented.
We thank the Reviewer for this comment. We have modified the Table (now Table 2) to include these suggestions.
In spite of greater understanding of molecular mechanisms of HCC pathogenesis, staging and treatment systems such as BCLC still based on morphological criteria to subdivide patients and direct management strategies. I would appreciate it if the authors were to provide some comment on the fact that there is ongoing need to refine treatment algorithms by including molecular markers indicative of high-, intermediate-, and low(er)-risk tumor biology, for example.
We agree with the Reviewer about the relevance of this suggestion. Therefore, at the end of the first section, when concluding about currently available therapies, we have included the following paragraph:
“These results suggest that HCC therapy is still far from being optimal, mainly for pa-tients at advanced stages, who mainly have a dismal prognosis, making necessary the development of new treatments with higher response rates and prolonged overall survival. Indeed, despite the greater understanding of molecular mechanisms of HCC pathogenesis attained during the last years, staging and treatment systems such as BCLC are still based on morphological criteria to subdivide patients and direct management strategies. Therefore, there is an ongoing need to refine treatment algorithms by including molecular markers indicative of high-, intermediate-, and low-risk tumor biology.
In Figure 1, the authors should differentiate “non-tumoral” liver from HCC, as the liver in the image is clearly not healthy. If they are truly aiming to differentiate healthy liver from a liver with chronic liver disease and HCC, then they should modify the image accordingly. Also, image “A” really adds nothing to the understanding of the readership (ball of cells representing a tumor, superimposed on the image of a “healthy liver”) and can be removed from the manuscript.
We agree with the Reviewer that panel A adds nothing to the reader and that it does not represent a healthy liver. Therefore, it has been eliminated and the legend modified accordingly.
Reviewer 3 Report
Dear Authors,
In my opinion, this is well written article, with use of very nice English language, with a need of only a few minor editorial reconsiderations (example: immune suppressive or immune-suppressive immunosuppressive ???) ... The subject is extremely important and it is also one of the most hot topics for HCC specialists ... As a reviewer, I strongly recommend this paper for publication. I congratulate for this work, especially with a use of the most recent bibliography :)
Best Regards :)
MW
Author Response
In my opinion, this is well written article, with use of very nice English language, with a need of only a few minor editorial reconsiderations (example: immune suppressive or immune-suppressive immunosuppressive ???) ... The subject is extremely important and it is also one of the most hot topics for HCC specialists ... As a reviewer, I strongly recommend this paper for publication. I congratulate for this work, especially with a use of the most recent bibliography.
We thank the reviewer for this comment, and following these suggestions we have modified the terms using “immunosuppressive” in all cases.